# The role of emergency departments in opioid related harms: A qualitative study among emergency healthcare providers

Charlotte Ten Pas[1]☺*, Viren Bahadoer[2‡], Cornelis Kramers[3‡], Albert Dahan[4‡¤],
Hannah Ellerbroek[5,6‡], Joris Holkenborg[7‡], Ozcan Sir[8‡], Merel Van Loon[9‡],
Arnt Schellekens[5,6‡], Nicole Kraaijvanger[2☺]

**1** Department of Emergency Medicine, OLVG, Amsterdam, Netherlands, **2** Department of Emergency
Medicine, Leiden University Medical Center, Leiden, The Netherlands, **3** Department of Pharmacology-
Toxicology, Radboud University Medical Center, Nijmegen, Netherlands, **4** Department of Anaesthesiology,
Leiden University Medical Center, Leiden, Netherlands, **5** Department of Psychiatry, Radboud University
Medical Center, Nijmegen, Netherlands, **6** Nijmegen Institute for Science Practitioners in Addiction
(NISPA), Nijmegen, The Netherlands, **7** Department of Emergency Medicine, Rijnstate Hospital, Arnhem,
The Netherlands, **8** Department of Emergency Medicine, Radboud University Medical Center, Nijmegen,
Netherlands, **9** Department of Emergency Medicine, Haaglanden Medical Center, The Hague, Netherlands

☺ These authors contributed equally to this work.
‡ VB, CK, AD, HE,JH,OS and MVL,AS also contributed equally to this work.
¤ Current address: Centre for Human Drug Research, Leiden, The Netherlands
* lottemaud@hotmail.com

journal.pone.0338421

of Technology, NEW ZEALAND

**Peer Review History:** PLOS recognizes the
benefits of transparency in the peer review
process; therefore, we enable the publication
of all of the content of peer review and
author responses alongside final, published
articles. The editorial history of this article is
available here: https://doi.org/10.1371/journal.
pone.0338421

## Abstract

### Objectives

Opioids are commonly prescribed in the Emergency Department (ED) for acute pain
management. However, their use carries significant risks, including dependence and
misuse. This study aims to gain insight into the perspectives of ED physicians and
physician assistants (PAs) concerning the multifaceted role of the ED in problematic
opioid use. This is crucial for reducing opioid-related harms.

### Design

A qualitative study, using semi-structured interviews.

### Setting and participants

Interviews were conducted with twenty-five ED physicians and PAs from four hospi-
tals in the Netherlands. A diverse group participated in the study, representing differ-
ent hospital settings and levels of working experience,

### Outcome measures and analysis

Reflexive thematic analysis was performed to develop key themes reflecting partici-
pants' perspectives and attitudes towards the role of the ED in opioid related harms.

**Data availability statement:** Data Availability Statement: Data is not publicly available due to ethical and legal restrictions related to participant confidentiality. The data contain sensitive information from qualitative interviews with individuals, and public sharing would compromise participant privacy. The data are held by Leiden University Medical Center (LUMC), and additional data can be requested from LUMC Division 1 Secretariat via email at secretar-iaatbbd1enzorg@lumc.nl. All requests will be reviewed by the institutional ethics committee.

**Funding:** This work was partially funded by Spoedeisende Geneeskunde Onderzoeksfonds via a 'High Potential Grant' awarded to CTP. The funder had no role in study design, data collection and analysis, decision to publish, or preparation of the manuscript.

**Competing interests:** The authors have declared that no competing interests exist.

## Results

Two key themes were developed from the analysis. The first theme 'Preventing opioid-related harms from an ED perspective' underscores the careful approach emergency physicians take when prescribing opioids. This involves restricting opioid prescriptions to specific indications, considering alternative pain management options, limiting prescription durations, and involving patients in shared decision-making. Beyond their own prescribing practices, emergency care providers also collaborate with general practitioners, navigate patient expectations, and operate within a broader societal landscape where pain is increasingly viewed as intolerable. The second theme 'Managing problematic opioid use at the ED' highlights the difficulties faced by emergency care providers in treating individuals who are already using or dependent on opioids. This includes recognizing, intervening, referring, and managing cases of problematic opioid use. This theme also considers the involvement of other healthcare professionals, such as pain specialists and psychiatrists, as well as the roles and responsibilities of patients. Additionally, it considers the broader societal context, particularly the extent of opioid-related harms in the Netherlands,

## Conclusions

This study sheds light on the complexities surrounding opioid use and emergency care providers' approach to mitigating opioid related harms while navigating patient needs and systemic challenges. EDs play a critical role in addressing opioid-related harms but face significant challenges. Strengthening provider education, integrating patient records, and enhancing partnerships with addiction services are key steps toward refining healthcare responses and policies for this ongoing public health crisis.

## Introduction

Opioids are potent analgesics frequently prescribed for acute pain management in emergency departments (EDs). However, opioid use carries significant health risks, including opioid-related harms, such as overuse and dependency, addiction, and overdose (fatal and non-fatal). Beyond these direct harms, the discontinuation of prescription opioids can lead to diversion toward illicit street opioids, increasing exposure to unsafe or adulterated drug supplies and associated injection drug use risks (HIV, hepatitis B/C, abscesses.) The United States (US) is currently facing an opioid abuse epidemic, including widespread dependency and high mortality rates [1]. The "opioid epidemic" was initially driven by the use of prescribed opioids and more recently by the use of illicit opioids, particularly fentanyl [1,2]. In Europe, most countries have also seen an increase in opioid prescriptions in the last decades together with associated hospital admissions [3]. In the Netherlands, prescription opioid use nearly doubled between 2008 and 2017 and opioid-related hospital admissions increased from approximately 2,5–7,8 per 100,000 inhabitants in that period [4]. The increased

use of prescribed opioids in Europe has been paralleled by a rise in opioid-related deaths in several European countries, including the Netherlands [5,6]. Overall, opioid-related harms have yet remained more contained in the Netherlands, partly due to quantifiable differences in healthcare organisation, advertising regulations and prescribing practices [4,6].

The Netherlands' universal healthcare coverage, high population density, and swift access to emergency services may help prevent the development of opioid misuse and overdose by facilitating early intervention and comprehensive care. The healthcare system is designed to ensure equal access for all citizens, offering comprehensive insurance in a regulated market that includes free primary care. Every Dutch citizen is registered with a general physician (GP), who is typically the first point of contact for health problems. Additionally, after-hours care is managed through large-scale, uniform GP cooperatives that cover 99% of the population. These cooperatives are available from 5 p.m. to 8 a.m. on weekdays and around the clock on weekends, using a nationally uniform model with telephone triage and physicians always present on-site [7]. In contrast, the US healthcare system is characterized by greater fragmentation, variable access to care, and less centralized oversight [8].

The Netherlands and most European countries have strict regulations prohibiting direct-to-consumer advertising of prescription medications and tight controls on pharmaceutical marketing to prescribers [4]. The Netherlands also has centralized and regulated dispensing, with most opioids prescribed in general practice and under stricter monitoring [9]. In contrast, the US healthcare system has allowed marketing and direct-to-consumer pharmaceutical advertising by pharmaceutical companies, which has been thought to have contributed to higher opioid prescribing and opioid-related harms [10,11]. Historically, in the Netherlands, opioid prescriptions are generally characterized by lower dosages and a preference for non-opioid analgesics within multimodal pain management [9,12]. Conversely, in the US opioids have been prescribed at higher doses and for longer durations, with a preference for potent opioids such as oxycodone as first-line therapy [13].

Because opioids are commonly prescribed at the ED, prescribing practices at EDs may contribute to opioid-related harms. Several studies have indeed shown that opioid prescriptions from EDs could lead to subsequent opioid misuse and to the development of opioid use disorder (OUD) [14,15]. A Canadian survey among emergency physicians found that they often do not perform a formal risk assessment before prescribing opioids but rely on intuitive estimation rather than identifying literature-based risk factors [16]. A previous study also showed that physicians' perceptions regarding opioids influence their prescribing behaviour, potentially leading to overprescribing, increasing the risk of opioid addiction, or under-prescribing, possibly resulting in inadequate pain management [17,18]. In response to rising opioid misuse, several countries have implemented monitoring systems to improve prescribing oversight. In many US states, prescribers are required by law to consult the Prescription Drug Monitoring Program (PDMP) before issuing an opioid prescription. Comparable national initiatives are largely absent in Europe, where efforts to reduce opioid-related harms have focused more on clinical guidance and professional awareness rather than prescription surveillance [9].

In the Netherlands, the Dutch Society of Emergency Physicians (DSEP) does not provide specific guidelines regarding pain management at the time of ED discharge. However, the Netherlands Society for Anesthesiology (NSA) offers a 'Generic Guideline Module for Appropriate Opioid Use,' which targets secondary healthcare settings, including EDs [19]. This guideline advocates for a maximum prescription duration of seven days and favors short-acting opioids. A recent study found that out of 33 EDs in the Netherlands, 58% follow locally developed opioid prescribing guidelines, which generally recommend short-term use of 3–7 days. Only 11% adhere to the national Dutch guideline, while the rest do not use any formal guideline for prescribing opioids at discharge. Around 40% of these local guidelines lack specific patient education, and when provided, mainly cover side effects and rarely mention addiction risks [20]. In contrast, following the severe opioid crisis, in the US more comprehensive guidelines have been developed for opioid prescribing in EDs. Recent guidelines from the Centers for Disease Control (CDC) and the American College of Emergency Physicians (ACEP) suggest a 'short duration' of opioid therapy, emphasizing clinical judgement and shared decision-making over strict limits on prescription duration [21,22].

Further, EDs encounter a range of opioid-related harms, including opioid seeking, withdrawal, overdoses, and injection-related complications. A recent Dutch study reported that 15% of patients presenting to the ED were actively using prescription opioids; among them, 23% showed signs of misuse and 10% met the diagnostic criteria for opioid use disorder (OUD) [15]. Presentations related to prescription opioid misuse or OUD could provide a window of opportunity for intervention in those patients who appear to be dependent on opioids [3,15]. For instance, in the US, emergency physicians are increasingly involved in prescribing buprenorphine to relieve acute withdrawal symptoms and initiate treatment for OUD [23]. However, a US-based mixed-methods study revealed that emergency physicians often experience personal frustration, helplessness, sadness, and dissatisfaction when dealing with patients with OUD in the ED. Their management options were often felt to be inadequate and limited [24]. A qualitative study from Canada highlighted the challenges faced by emergency physicians in managing OUD and opioid withdrawal. The study underscored concerns regarding follow-up care while managing withdrawal, the challenges of prescribing long-term medications such as buprenorphine in an acute care setting, and patient attitudes towards detoxification [25].

The Netherlands, with its universal health insurance and regulatory oversight, has so far succeeded in keeping opioid-related harms relatively contained. Although many European countries share similar regulatory safeguards, such as bans on direct-to-consumer pharmaceutical advertising, they often lack the Netherlands' highly integrated primary care infrastructure. Lessons for other European healthcare systems include prioritizing robust primary care gatekeeping, implementing regular medication reviews, developing practical prescribing tools, and fostering interdisciplinary collaboration [26,27]. However, rising trends in opioid prescriptions, hospital admissions, and mortality within the Netherlands highlight the need for ongoing vigilance. Strengthening the role of primary care in overseeing safe pain management remains essential, both nationally and across European systems seeking to address emerging opioid concerns. At the same time, differences in healthcare organization, prescribing culture, and national guidelines may limit direct generalizability to other European countries.

In conclusion, although both the Netherlands and the US face challenges related to opioid misuse, they differ markedly in healthcare systems, prescribing practices, and regulatory frameworks. While opioid-related harms in the Netherlands remain relatively contained, prescription use, hospitalizations, and mortality have risen sharply in recent decades. EDs serve as critical points of care for acute opioid-related presentations and offer valuable opportunities for intervention. The primary objective of this study is to investigate the perceptions of ED physicians and PAs concerning the multifaceted role of emergency healthcare providers in opioid-related harms in the Netherlands. This understanding forms the foundation for developing targeted interventions, ensuring that guidelines for ED physicians and educational initiatives are tailored to align with these insights. Until now, no qualitative research on this topic has been conducted in Europe. Given the different characteristics of the opioid situation, healthcare organization, and social support systems in the Netherlands, it is of interest to add this perspective.

## Methods

### Study design and setting

This qualitative interview study employed semi-structured interviews to investigate the perceptions of physicians and physician assistants (PAs) regarding opioid prescribing and opioid-related harms in the ED setting. The study was conducted with a purposive sample of PAs, residents, and emergency physicians working in various (academic and non-academic) representative EDs, located in both urbanized and less urban regions in the Netherlands: Leiden University Medical Center (LUMC, an academic center with 25,000 ED visits annually), Haaglanden Medical Center Westeinde (HMC, a teaching hospital and level-I trauma center with 55,000 ED visits annually), Radboud University Medical Center (RadboudUMC, an academic level-I trauma center with 16,000 ED visits annually), and Rijnstate Hospital (a teaching hospital with 35,000 ED visits annually). Small rural hospitals were not included in our sample. However, this type of hospitals is scarce within the

Dutch healthcare system. The sample thus offers a strong reflection of current practice and captures the diversity most relevant to the national context.

## Participants

Participants were health care providers working in the ED, with the authority to prescribe opioids. Purposive sampling was utilized to select a diverse group of healthcare professionals, considering factors such as age, sex, profession, and years of experience, while also achieving variation across hospital types and regions. This was done to provide a comprehensive perspective on opioid prescribing, use, and misuse within the ED context. An invitation to participate in the study was emailed to all emergency physicians and PAs working in the four participating EDs (around 20 clinicians per site) several weeks prior to the interviews. In addition, the interviewer visited each ED on multiple occasions to facilitate recruitment. No financial or other incentives were offered. Nearly all clinicians who were approached agreed to participate; the few who did not participate cited scheduling conflicts or workload as the main reasons. Participants understood that their statements might be quoted in the manuscript. The participants did not have the opportunity to provide feedback on the study findings.

## Interviewer role

All interviews were conducted by C.t.P., a resident physician with two years of working experience in the ED. The interviewer was known within the LUMC but had no prior relationship with participants from the other three hospitals. Participants were informed that the interviewer was a resident at the LUMC ED and that the study aimed to explore physicians' perspectives on opioid prescribing and opioid-related issues. They had the possibility to ask additional questions before the start of the interviews, and were assured that their input would be used solely to improve understanding of the topic. The interviewer's background in emergency care enriched the credibility and quality of the insights gathered. As participants interacted with someone who understood the complexity and nuances of their work environment, they were more inclined to engage in detailed discussions about their experiences and challenges.

## Data collection

Data were collected through semi-structured interviews. The interviews were conducted using a topic guide developed for this study by all authors based upon previous literature and experience [19]. The topic guide was designed to elicit detailed responses regarding participants' views on opioid prescribing, their experiences and attitudes regarding opioid-related harms in the ED, allowing participants to elaborate freely on their perspectives. Follow-up and probing questions were asked to clarify or deepen responses when relevant. Data saturation was achieved after 20 interviews, as no new additional significant information was captured beyond this point. To ensure saturation, five additional interviews were conducted using purposive sampling, also ensuring diversity. These interviews did not provide relevant new insights. The 25 interviews were all conducted in person by C.t.P, a resident in emergency medicine between January 2024 and May 2024.

In addition to questions regarding opioids, participant demographic information (age, gender, and work experience) was collected. These data were stored securely and retained for up to one year following data collection.

## Data analysis

The interviews were audio-recorded, transcribed verbatim (using Amberscript transcription software). To develop a comprehensive understanding of each physician's experience, the transcripts were read and reread thoroughly. Two researchers C.t.P. (female emergency medicine resident with no prior experience in qualitative research) and N.K., PhD, emergency physician, experienced in qualitative research) independently coded the transcripts with ATLAS.ti software (Version 23.2.3.27778) using the reflexive thematic analysis approach as described by Braun & Clarke [28]. This involved

identifying inductive descriptive codes by highlighting recurring phrases within the physicians' narratives. Following the initial coding, C.t.P. and N.K. compared and discussed the codes, thereby enhancing the depth of the analysis. The codes were organized into themes over the course of two thematization meetings. In these sessions, the codes were collaboratively grouped to construct themes that reflected the content and nuances of the interview data. The first meeting took place after the first 15 interviews were conducted and included researchers C.t.P., N.K., A.S. (professor in addiction psychiatry, experienced with qualitative study methods), H.E. (PhD candidate with a background in qualitative research on opioid misuse) and V.B. (medical student/ PhD candidate in research on opioid misuse). The topic guide was refined during this initial meeting, adding questions about personal experiences with opioid-related harms to deepen the discussions. The second session was held after all 25 interviews were completed and involved C.t.P., N.K., A.S. and H.E. The study results were reported in accordance with the Consolidated Criteria for Reporting Qualitative Research (COREQ) [29]. The manuscript is reviewed and corrected using AI-based tools, specifically for grammatical accuracy and clarity.

## Ethical considerations

Ethical approval was obtained from the Leiden University Medical Center (LUMC) ethics committee (protocol identification number N052). The study was conducted in accordance with the ethical standards of the institutional and national law. All participants provided written informed consent before they participated in the study. Their anonymity and confidentiality were maintained throughout the research process.

## Results

A total of 25 participants was included in the study, with 9 recruited from RadboudUMC, 5 from Rijnstate Hospital, 8 from HMC, and 3 from LUMC. The interviews were all conducted in person at the different EDs between January 2024 and April 2024. The median length of the interviews was 28 minutes, ranging from 15 to 42 minutes.

### Characteristics of participants

The interviewees comprised thirteen women (52%) and twelve men (48%) and included twelve emergency physicians, ten resident physicians from various specialties (from which two emergency residents) and three physician assistants.

The average age of the participants was 35 years, ranging from 25 to 59 years. On average, the participants had ten years of experience in their current roles (median of four years of experience ranging from six months to 27 years) and on average eight years of experience in the ED (median of five years, range of several months to 27 years).

Twelve out of twenty-five (48%) had previously worked at another ED, and seven (28%) participants had prior clinical experience outside the Netherlands, mainly within Europe, and a few in the US. Additionally, participants had backgrounds in various specialties, including Surgery (20%), Internal medicine (16%), Neurology (16%) and Cardiology (16%), Intensive Care (12%), among others. Detailed participant information is provided in Table 1.

### Themes

Through reflexive qualitative analysis, two main themes were developed, each encompassing three subthemes, see Table 2. These themes captured the perceptions and experiences of emergency care providers in opioid prescribing and addressing opioid-related harms at the ED.

### Theme 1: Preventing opioid-related harms

This theme addresses the prevention of opioid-related harms, through thoughtful and conservative prescribing of opioids at the ED. In three subthemes, the roles of the involved healthcare providers, patients and the social context are discussed.

**Table 1. Baseline demographics and characteristics of the participants.**

| Characteristics participants | Total (n=25) |
|---|---|
| Sex, n (%) | |
| Male, | 12 (48%) |
| Female | 13 (52%) |
| Age, mean (SD) | 38.6 (9.9) |
| Current role, n (%) | |
| Emergency Physician (EP) | 12 (48%) |
| Physician Assistant (PA) | 3 (12%) |
| Junior Doctor in Surgery | 3 (12%) |
| General Practitioner in Training | 2 (8%) |
| Junior Doctor in Emergency Medicine | 2 (8%) |
| Resident Surgery | 1 (4%) |
| Resident Geriatrics | 1 (4%) |
| Resident Internal Medicine | 1 (4%) |
| Working experience in current role in years (SD) | 8.0 (7.8) |
| Total working experience at ED, years (SD) | 9.0 (8.7) |
| Working experience at another ED, n (%) | 12 (48) |
| Foreign working experience, n (%) | |
| Yes | 6 (24%) |
| Working experience besides ED at | |
| Internal medicine | 4 (16%) |
| Neurology | 4 (16%) |
| Cardiology | 4 (16%) |
| Pediatrics | 3 (12%) |
| Intensive care | 3 (12%) |

### Subtheme 1: Role of health care providers

**Responsibilities in prescribing opioids.** Several interviewed physicians highlighted that the ED often is the initial place for patients receiving opioid prescriptions and possibly a starting point of developing dependence. Many ED physicians were committed to be cautious in opioid prescribing.

Participant#4: *"The ED is the place where many patients come in with conditions that lead to an opioid prescription—fractures, abdominal pain, and so on. As a result, we are frequently the ones to write that initial prescription for these patients."*

Some emphasized that a risk assessment for addiction should be conducted before any prescription is issued. Further, the consensus was that the necessity for opioid prescriptions should be based on clinical assessment and judgment by physicians or PAs.

Administering an opioid within the ED was perceived as very different from providing a take-home prescription, with opioid administration in the ED being done more readily than prescribing them for home use. An often-mentioned challenge with take-home prescriptions was the lack of follow-up once the patient leaves the ED. Several participants expressed discomfort about this discontinuity, noting that it left them uncertain about whether their actions had contributed to ongoing opioid use or dependency.

**Table 2.** Overview of main themes and subthemes, including illustrative quotes.

| Main Theme | Subtheme | Descriptions | Illustrative Quote |
|---|---|---|---|
| 1. Preventing opioid related harms | Healthcare providers | Responsibilities in pre-scribing opioids | "The ED is the place where many patients come in with conditions that lead to an opi-oid prescription—fractures, abdominal pain, and so on. As a result, we are frequently the ones to write that initial prescription for these patients." |
| | | Opioid prescriptions at ED discharge | "Here, we strictly prescribe opioids for a maximum of five days, and usually only for three days, without allowing the prescription to be renewed. I believe this approach significantly reduces the risk of developing an addiction." |
| | | Shared decision making in the ED | "You want to make a meaningful difference for the patient, and I believe this should be done through shared decision-making. This involves engaging the patient, acknowl-edging their knowledge and perspective, while also helping them understand the potential negative effects of opioids. The conversation is key." |
| | | Perceived roles of other healthcare providers | "GPs know their patients better than we do in the ED, and that allows them to make more nuanced decisions about pain management." |
| | Patients | Knowledge and expecta-tions regarding opioids | "There are always high expectations from patients: 'Doctor, I'm in pain, or I have this problem,' and they expect you to fix it. The easiest solution is often to prescribe medication." |
| | Social context | Acute nature of emer-gency care | ''Doctors in general find it difficult to say no to patients, whether it concerns additional diagnostics or pain management. Since we have something to address the pain, opioids, we tend to prescribe them." |
| | | Pain in society | ''There seems to be a trend where opioids are prescribed quite readily in the ED, as if pain is no longer considered acceptable." |
| | | Responsible prescribing with EPF | ''I think it would be beneficial to have a more unified, nationwide EPF and shared insights across EDs. Currently, there is no way to share data between facilities. For example, if a patient visits one ED and later goes to another, there's no way of knowing." |
| 2. Managing opioid misuse and OUD in the ED | Healthcare providers | Dealing with opioid mis-use and OUD in the ED | "We have a strong role in identifying problematic opioid use! While treatment is more challenging and requires continuity, I believe we are well-positioned to fulfil a key role in early detection." |
| | | Starting OUD treatment at the ED | "If you develop a solid protocol for it, I think it's entirely feasible. They do it in the United States, so why couldn't we? The fact that we never do it here is a bad reason not to start. It requires experience, but more importantly, a well-structured protocol and a strong network." |
| | | Perceived roles of other healthcare providers | "We need more accessible ways to discuss patients at risk of addiction with GPs, so we can intervene early and make sure they get the right follow-up care." |
| | Patients | Patient responsibility in seeking help for OUD | "I can only advise patients to contact addiction services, but it has to come from their own willingness to do so. You can't force someone into addiction care—they really need to want it themselves and have to be motivated to take that step." |
| | | Opioid shopping and aggression | "Shoppers sometimes cause chaos in the ED, even leading to aggressive incidents. This creates a lot of stress and has a significant impact on the team, which may result in the perception that these individuals are beyond help." |
| | Social context | The extend of opioid-related harms in the Netherlands | "I think it's already quite a problem, but I don't believe we've reached the peak yet. I'm worried about it." |
| | | Perceived societal causes of opioid related problems | "There isn't a single cause for opioid-related problems; it's a combination of factors. It likely involves both nature and nurture—genetic predisposition, upbringing, social background, personality traits, potential personality disorders, coping mechanisms, the availability of addictive medications, and perhaps even the decline of the social involvement in society." |
| | | Shared EPF for opioid problem detection | "If someone visits this ED and later goes to another, we have no way of knowing. I think that makes it harder to implement effective interventions." |

Abbreviations: *ED = Emergency Department, EPF = Electronic Patient File, OUD = Opioid Use Disorder*

*Participant #10: "After the ED visit, I have no visibility on my patients anymore. I don't get feedback on the actions I took, and I have no idea about their follow-up."*

The importance of providing a clear and timely handover to the patient's general practitioner (GP) was repeatedly mentioned.

**Opioid prescriptions at ED discharge**

**Opioid prescription specifics.** For most participants, an aspect of responsible opioid prescribing at ED-discharge was a short duration of the prescription. For some this was 3–5 days, while others advocated for 1–2 days. One individual stated that addiction only develops with prolonged use, so opioid prescribing for a short duration should be no problem.

*Participant #16: "Here, we strictly prescribe opioids for a maximum of five days, and usually only for three days, without allowing the prescription to be renewed. I believe this approach significantly reduces the risk of developing an addiction."*

Physicians and PAs established that there is no clear protocol for prescribing opioids at the ED. Currently, just over half of Dutch EDs follow prescribing guidelines for pain management after discharge. Most rely on locally developed protocols, but these guidelines usually lack condition-specific recommendations [20].

There was consensus that opioids should not be prescribed for chronic pain and that prevention of addiction is a priority when prescribing opioids. Opinions about prescribing long-acting or short-acting preparations varied. Some participants expressed a preference for long-acting opioids, perceiving them to be less addictive than short-acting ones. Conversely, other participants held the opposite view.

**Indications and contraindications.** Generally, physicians and PAs emphasized pain medication should be tailored to individual patients, considering factors such as prior substance use, pain perception, age, comorbidities, and home pain management needs.

For acute painful conditions, such as traumatic injuries (fractures), postoperative pain, or severe abdominal pain (e.g., renal stone colic), opioids were generally seen as appropriate analgesia.

*Participant #12: "Acute pain becomes a problem if it's not managed properly. I believe there's significant benefit in training clinicians to always address it and incorporate it into the primary survey. Additionally, increasing awareness of how harmful untreated pain can be is equally important."*

However, in unspecified abdominal pain, back complaints or chronic pain, most physicians and PAs avoided opioids. A history of addiction or previous chronic opioid use was considered as a contraindication for opioids. However, participants noted that this history is not always explored in practice, often due to time constraints in the emergency setting or perceived lack of relevance. One participant specifically identified a history of bariatric surgery as being a risk for problematic opioid use. Further, a greater risk of side effects in elderly patients (e.g., confusion and falls) was mentioned to be a contraindication for opioids.

Yet, several participants expressed that inadequate analgesia can also lead to complications and an increased risk of chronic pain, and that proper management of acute pain at the ED is crucial. One participant highlighted the pain score as a fifth vital sign, underscoring the importance of adequately addressing pain. On the contrary, a few physicians noted that pain scores alone should not be the decisive factor in selecting appropriate analgesic therapy.

**Alternative pain medication.** According to most physicians, efforts are made to provide pain relief with other medication before opioids are prescribed, with paracetamol and NSAIDs as the first options. Participants reported to be reluctant to prescribe NSAIDs because of possible side effects and interactions, yet they were generally seen as a useful

alternative to opioids. Tramadol was sometimes seen as a middle ground between NSAIDs and strong opioids (e.g., oxycodone) and considered especially when NSAIDs were not appropriate due to comorbidity or age.

**Shared decision-making**

Communication with patients regarding their preferences and opioid-associated risks were often emphasized. Some physicians mentioned that patients often forget the provided information, and may use more opioids than prescribed, which can contribute to development of dependence. Several participants mentioned that shared-decision making can be difficult due to time constraints in the ED.

> Participant #14: *"You want to make a meaningful difference for the patient, and I believe this should be done through shared decision-making. This involves engaging the patient, acknowledging their knowledge and perspective, while also helping them understand the potential negative effects of opioids. The conversation is key."*

**Perceived roles of other healthcare providers**

**The role of general practitioners.** Some ED physicians pointed out that opioids are prescribed more easily in the ED, where acute pain is more prevalent, compared to GPs and after-hours GP-cooperatives, who are considered more conservative with prescribing pain medication. However, several ED physicians mentioned that GPs may sometimes be tempted to prescribe opioids to maintain a positive relationship with their patients.

> Participant #4: *"GPs know their patients better than we do in the ED, and that allows them to make more nuanced decisions about pain management."*

**The role of medical specialists.** Some ED physicians perceived that surgical departments frequently discharge patients with opioid prescriptions with a longer duration compared to those issued by the ED.

> Participant #1: *"Everyone who underwent surgery performed by the surgical department was given dual opioid prescriptions, consisting of short-acting and long-acting oxycodone, for a duration of two weeks."*

**Subtheme 2: Patients**

**Knowledge and expectations regarding opioids.** Participants noticed that patients often have limited knowledge about the correct use and associated risks of taking prescription opioids. Some participants noted that patients were unaware of which opioids they were taking, what the adverse effects may be and the fact that opioids are addictive. Several physicians found that patients generally have poor knowledge of pain management, adding to the risk of misusing painkillers.

> Participant #10: *"Some patients are surprised when I inform them they will receive a morphine tablet, but I do so intentionally to ensure they are aware of what they are taking and to encourage thoughtful use. The goal is not to create a barrier but to emphasize that pain relief is essential and the medication should be used when genuinely needed. However, I also want to caution them against taking it simply because it is readily available at home."*

Interviewed physicians observed that some patients, even those without prior opioid experience, actively requested opioids, perceiving them as the most effective solution for severe pain. Conversely, they also highlighted patients who refuse opioids due to concerns about addiction, side effects, or societal stigma.

> Participant #3: *"There are always high expectations from patients: 'Doctor, I'm in pain, or I have this problem,' and they expect you to fix it. The easiest solution is often to prescribe medication."*

### Subtheme 3: Social context

**Acute nature of emergency care.** Some physicians noted that they tended to prescribe opioids liberally due to the acute nature of the ED and the limited time available. Patients often present at the ED with severe pain, and there is a strong desire to provide relief.

Participant #5: ''*Doctors in general find it difficult to say no to patients, whether it concerns additional diagnostics or pain management. Since we have something to address the pain, opioids, we tend to prescribe them.*''

### Pain in society

Several participants expressed the perception that contemporary society often does not accept the experience of pain, leading to a strong desire to treat it.

Participant #1: ''*There seems to be a trend where opioids are prescribed quite readily in the ED, as if pain is no longer considered acceptable.*''

Conversely, some physicians noted a cultural tendency in the Netherlands to inadequately assess and treat pain in hospital settings, resulting in the undertreatment of pain. One of the physicians mentioned a reluctance to prescribe opioids due to the hospital's conservative opioid prescribing policy. Participants described ambivalent concerning two contrasting cultural perspectives on pain management. Some emphasized a traditional, more stoic approach to pain, characterized by the belief that pain is something to be endured rather than eliminated. This may contribute to undertreatment. Others highlighted a more recent shift toward the idea that pain should not be accepted and must be actively treated, supported by initiatives such as including pain as a (fifth) vital sign in triage protocols. This may contribute to overtreatment. These contrasting tendencies coexist within Dutch emergency medicine, reflecting broader societal and professional debates about what constitutes appropriate pain management.

### Responsible prescribing with Enhanced Electronic Patient File (EPF)

Several physicians highlighted the lack of oversight and coordination in prescribing practices between hospitals, making it challenging to track what other providers may have already prescribed. This gap increases the risk of duplicate prescriptions and facilitates "opioid shopping". Physicians and PAs expressed a strong need for improved information sharing.

Participant #3: ''*I think it would be beneficial to have a more unified, nationwide EPF and shared insights across EDs. Currently, there is no way to share data between facilities. For example, if a patient visits one ED and later goes to another, there's no way of knowing.*''

### Theme 2: Managing opioid misuse and OUD in the ED

In this theme the focus is on addressing the challenges in the ED in patients who are already using or dependent on opioids. Again, in three subthemes, the roles of the involved healthcare providers, patients and the social context are considered, see Table 2.

### Subtheme 1: Role of health care providers

**Dealing with opioid misuse and OUD at the ED.** *Treating pain in patients already using opioids*
The interviewed healthcare providers reported facing a dilemma when deciding whether to prescribe additional opioids for patients who are in pain but already are using opioids, and for those with suspected opioid dependence. Some expressed concern that prescribing more opioids could exacerbate a patient's dependency on opioids.

Participant #3: *"When someone is in pain, we feel the need to address it with medication... but I don't think that's always necessary. Pain can often be managed through other approaches, like coping strategies. Higher doses of pain medication are not always the solution, nor do the benefits always outweigh the risks. This is something you can and should discuss with the patient."*

In cases where problematic opioid use was suspected, most interviewed physicians preferred prescribing a limited opioid supply, followed by referral to the patient's GP. Other possible solutions mentioned by the physicians included opioid rotation, locoregional anesthetics, or addressing pain through psychological support and coping strategies.

**Recognizing problematic opioid use.** Several healthcare providers had difficulties identifying problematic opioid use in patients.

Participant #7: *"We have a strong role in identifying problematic opioid use! While treatment is more challenging and requires continuity, I believe we are well-positioned to fulfil a key role in early detection."*

They emphasized the importance of inquiring about patients' home use of pain medication. A patient asking for opioids outside one's local area was generally considered a red flag, signalling potential misuse or abuse. Suspicions of dependency were frequently discussed with nurses and other healthcare providers. The process of detecting addiction and responding effectively was often considered challenging and time-consuming by the participants, but also because of the stigma surrounding substance use.

Participant #10: *"If you suspect problematic opioid use, then I believe it is also the responsibility of emergency medicine, as difficult as that may be. Not every ED physician or PA will appreciate this because it means you have to take the time, sit down with people, and that can take up an hour of your time."*

**Responsibilities of emergency healthcare professionals in addressing problematic opioid use.** Opinions differed on whether addressing prescription OUD falls within the scope of the ED. Some believed that the primary role of the ED is to prevent harm by avoiding or restricting opioid prescriptions, rather than intervening in an already present OUD. They mentioned time constraints, lack of insight in medical histories, and limited follow-up options as factors hindering the ED's ability to intervene when problematic opioid use is suspected. Some also noted that a history of substance use was often not explicitly explored during ED encounters, reflecting a certain discomfort or taboo around the topic.

Participant #10: *"I can't make promises or set up a follow-up plan. What I can do is start the conversation, build trust, and guide patients into the healthcare system. But beyond that, I have nothing to offer them. The moment they leave the ED, I lose them, and I know no one else will follow up. At best, I can mention it in the letter to the GP."*

Discussing suspected problematic opioid use with patients was mentioned several times as complex. Others argued that opioid addiction should be identified at the ED, as it is often the first point of contact for patients. They saw the ED-visit as an opportunity to address opioid-related harms during acute crises, initiating conversations about addiction and potential next steps, such as referrals to primary care or addiction services. However, treating addiction was generally viewed as beyond the ED's capacity.

Participant #5: *"The problem is that prescribing an opioid is often the easiest solution, and patients know this. They don't come in the morning when it's quiet. We are not the ideal place to make a well-considered decision or to provide substantial support to patients struggling with addiction. However, we do play a critical role in identifying these issues."*

**Starting OUD treatment at the ED**

**Referral to addiction treatment from the ED.** ED physicians generally expressed that developing a follow-up plan for patients with opioid dependency is a significant challenge. Many felt there is a shortage of resources and expertise available to adequately manage and refer these patients.

Several participants stated that the lack of availability of addiction services, particularly outside regular hours, often leaves patients without the support they need. ED staff frequently felt compelled to discharge opioid-dependent patients with opioid prescriptions due to the absence of better options. This situation was often perceived as frustrating.

Participant #2: ''Honestly, I wouldn't even know how to refer someone to addiction care, apart from trying to motivate them to actually go. It really should be easier.''

Participant #11: "We don't have standardized protocols for withdrawal management, and working with psychiatry or primary care often feels like a struggle because they are only available during office hours and generally difficult to reach."

**Prescribing Methadone or buprenorphine at the ED.** Physicians and PAs recognized that methadone and buprenorphine can be used to treat OUD, but they viewed prescribing these medications as outside their expertise and responsibilities, as the ED provides only short-term treatment.

Participant #4: "I don't think so. We have a very short treatment relationship, and I believe that (prescribing methadone) falls somewhat outside the scope of the ED."

Although some would be willing to prescribe methadone to prevent acute withdrawal symptoms, others emphasized that this would only be justifiable if a comprehensive follow-up plan is in place. A minority of participants expressed no concerns about prescribing methadone, but they stressed the importance of receiving training prior to initiating this practice.

Participant #5: "If you develop a solid protocol for it, I think it's entirely feasible. They do it in the United States, so why couldn't we? The fact that we never do it here is a bad reason not to start. It requires experience, but more importantly, a well-structured protocol and a strong network."

**Perceived roles of other healthcare providers**

**The role of general practitioners.** Collaboration between the ED and GPs was seen as essential, especially when problematic opioid use was suspected. However, there was an opinion among ED physicians that communication between the ED and GPs sometimes falls short. In the Dutch healthcare system, general practitioners routinely receive electronic discharge letters from the ED. However, several participants noted that while these communications are reliably transmitted, the large volume of correspondence makes it difficult for GPs to read and process every letter in detail. This contributes to concerns about whether advice or follow-up recommendations are consistently acted upon.

ED physicians generally found that the responsibility for coordinating opioid prescribing and misuse ultimately lies with the patient's GP, as they know their patient best and have an overview of their medication history. However, it was also recognized that a collaborative effort is needed to adequately manage opioid-related harms. Several ED physicians reported that GPs not always have the resources or support needed to effectively address problematic opioid use.

Participant #17: "We need more accessible ways to discuss patients at risk of addiction with GPs, so we can intervene early and make sure they get the right follow-up care."

**The role of addiction care.** ED physicians often expressed that the accessibility of addiction care is limited. Physicians highlighted that addiction services seemed to be primarily focused on recreational drug use, with less attention for addiction due to prescribed opioids.

Several physicians wished for an accessible intake office for addiction treatment, where referrals could be made easily, and also outside regular working hours.

Participant #10: "*If there would be a better addiction service in place, I would refer more patients there. As it is now, addiction care often takes months to respond to a referral, which is completely ineffective.*"

*The role of medical specialists*

**Pain specialists.** Several physicians felt that it can be useful to involve pain specialists (anesthesiology) in managing pain and preventing complications of opioid use. A few physicians suggested follow-up with pain specialists or anesthesiology as a solution for better continuity of care for these patients.

Participant #7: "*Consulting the pain specialist at the ED or sending them to the outpatient clinic for follow-up seems like the best solution. They can be very helpful in managing chronic pain patients and addressing opioid issues.*"

**Psychiatry.** Several interviewed physicians believed there is a need for regular education and training to ED personal, led by addiction specialists and psychiatrists, to better inform and support decision-making around opioid prescriptions. They highlighted the value of involving experts in complex substance abuse cases, since they are better equipped to conduct conversations on addiction and guide appropriate interventions.

Participant #10: "*Psychiatry is essentially trained for these kinds of conversations. So, I can imagine that if we identify these issues, we should request psychiatry consultation.*"

**Subtheme 2: Patients**

**Patient responsibility in seeking help for OUD.** Physicians emphasized that while they can provide advice and referrals to addiction care, the patient must actively follow up on these recommendations. Many physicians agreed that addiction care inherently requires patient motivation for recovery, but they acknowledged that patients with addiction can have limited self-awareness, complicating this process.

Participant #6: "*I can only advise patients to contact addiction services, but it has to come from their own willingness to do so. You can't force someone into addiction care—they really need to want it themselves and have to be motivated to take that step.*"

Further, there was a shared sentiment that the current healthcare system provides insufficient support for patients struggling with opioid dependence.

**Opioid shopping and aggression.** Many physicians adopt a cautious approach to patients requesting opioids, viewing such requests as potential red flags for misuse or dependency. Some physicians noted that patients sometimes visit the ED outside office hours to obtain opioids more easily.

Participant #5: "*I often already know these patients because there are frequent flyers. In advance, the team decides whether to give no opioids or only a single dose for these patients. Having an open conversation with the patient also helps in this process.*"

These patients were often seen as demanding, especially when they resist alternative treatments or lack a clear indication for opioid use. Some participants described situations where patients became aggressive when their requests for opioids were denied, creating an unpleasant and unsafe feeling.

Participant #3: "*Shoppers sometimes cause chaos in the ED, even leading to aggressive incidents. This creates a lot of stress and has a significant impact on the team, which may result in the perception that these individuals are beyond help.*"

### Subtheme 3: Social context

**The extend of opioid-related harms in the Netherlands.** The interviewed physicians expressed diverse, sometimes contradictory opinions on the current extent of opioid-related harms in the Netherlands. There was acknowledgment that opioid abuse is an issue in the country. Additionally, some believed that the opioid problem may be more widespread than visible and that the peak of opioid-related harms in the Netherlands is yet to come.

Participant #4*: "I think it's already quite a problem, but I don't believe we've reached the peak yet. I'm worried about it."*

However, several physicians expressed that the opioid situation in the Netherlands is not comparable to the severe crisis in the United States. The Dutch healthcare system, with its stricter prescribing practices, is thought to contribute to a lower prevalence of opioid addiction.

Participant #6: "*Maybe I'm being naïve, but I don't have the impression that this is a big problem in the Netherlands.*"

**Perceived societal causes of opioid related problems.** The role of the pharmaceutical industry is debated. While some doctors see the industry as a key contributor to the problem, the influence of pharmaceutical companies in the Netherlands is considered to be smaller compared to the US. The problem of opioid addiction was described as multifactorial by several physicians, involving the prescriber, the patient, the pharmaceutical industry, and the broader societal tendency to view pain as something that must be eliminated.

Participant #23: "*There isn't a single cause for opioid-related problems; it's a combination of factors. It likely involves both nature and nurture—genetic predisposition, upbringing, social background, personality traits, potential personality disorders, coping mechanisms, the availability of addictive medications, and perhaps even the decline of the social involvement in society.*"

### Shared Electronic Patient Files (EPF) for opioid problem detection

Several interviewed physicians advocated for a nationally shared EPF system to provide insight into prescribed medications and improve care coordination. These systems could implement alerts for increased dosages and documenting opioid misuse.

Participant #3: "*If someone visits this ED and later goes to another, we have no way of knowing. I think that makes it harder to implement effective interventions.*"

Participant #21: "*I think we lack the tools to make a meaningful contribution to this problem on a societal level. For instance, having a nationwide EPF for the ED could help identify shoppers or patients who visit multiple EDs because of opioid intoxications. With such a system, we would have more ground to address the issue effectively and stronger grounds to refer patients or initiate conversations with them.*"

## Discussion

This qualitative study investigates the perspectives of emergency care providers regarding the role of the ED in opioid-related harms. Two main themes were developed from the analysis.

The first theme 'Preventing opioid-related harms from an ED perspective' highlights cautious and deliberate opioid prescribing practices within the ED. This includes limiting indications for opioid prescribing, exploring alternative pain management strategies, prescribing opioids for short durations, and engaging in shared decision-making processes with patients. Beyond the role of emergency healthcare providers, this theme also explores their collaboration with GPs, the expectations and demands of patients, and the societal context in which pain seems increasingly perceived as intolerable.

The second theme 'Managing problematic opioid use at the ED' focuses on the challenges faced by emergency healthcare providers in treating patients already using or dependent on opioids. This encompasses the identification, intervention, referral, and treatment of problematic opioid use. Additionally, this theme encompasses the roles of other healthcare providers, such as pain specialists and psychiatrists, alongside patients' responsibilities and behaviours. It also takes into account the societal context, including the extend of opioid-related harms in the Netherlands.

Physicians acknowledge the dangers of prolonged opioid use and aim to limit prescriptions, yet they often feel constrained in their choices. Patient expectations, time pressure, and the lack of immediate alternatives significantly impact prescribing behaviour. There is a strong need for standardized protocols and improved education to support responsible prescribing. Unlike in the US, where pain has been institutionalized as the "fifth vital sign," this concept is not embedded in routine clinical practice in the Netherlands. In Dutch EDs, pain is incorporated into the Manchester Triage System as a clinical discriminator used to assess urgency, rather than being classified as a formal vital sign [30]. Standardized pain scores are commonly used for patients able to self-report, but clinical judgment often guides management [30]. Participants described a growing awareness of the importance of pain management, yet also noted differing views: some emphasized the need for proactive pain relief, whereas others highlighted that not all pain can or should be eliminated. This ambivalence reflects ongoing debate in Dutch emergency medicine about balancing adequate analgesia with cautious opioid prescribing.

ED physicians experience a gap in monitoring patients' long-term opioid use, which affects their prescribing decisions. Physicians agree that the ED is a key setting for recognizing opioid-related harms and initiating the first steps of treatment. Collaboration between ED physicians, general practitioners, psychiatrists and pain specialists is considered essential to reduce problematic opioid use.

### Preventing opioid-related harms from an ED perspective

Our findings indicate that emergency care providers often experience a dual responsibility in preventing opioid-related harms. Cultural attitudes toward pain and addiction further complicate care delivery. On one hand, they need to manage acute pain effectively. A few physicians emphasized the importance of adequate pain treatment, stressing that this is crucial to prevent the development of chronic pain. There is ample evidence that effective early management of severe acute pain is important in preventing the development of chronic pain [31–33]. On the other hand, emergency care providers emphasize the importance of reducing the risk of opioid-related harms, such as misuse, addiction, and overdose, by prescribing opioids with caution and in minimal amounts [1,34]. We found evident variations in opioid prescribing practices across EDs between participants, particularly concerning the selection of long-acting versus short-acting formulations and prescription duration.

Cultural attitudes toward pain and opioid use differ markedly between the Netherlands and the US. In the US, aggressive pain management and pharmaceutical marketing contributed to high opioid consumption and misuse [35,36]. In contrast, European countries, including the Netherlands, have more restrictive prescribing cultures and greater emphasis on non-opioid pain management [37].

Additionally, we found that patient expectations contribute to this dilemma. Patients in acute pain often require quick and effective relief, which can pressure providers to prescribe strong painkillers. Most physicians balance this dilemma by prescribing only short courses of opioid treatment. Currently, the treatment of acute pain still appears to depend heavily on opioids, as other therapeutic options for managing this pain are perceived to be limited. Some physicians emphasize the need for well-defined, evidence-based protocols on opioid prescribing in acute care settings to support their decision-making processes. Notably, our findings on opioid prescribing at ED discharge are largely consistent with those reported in a qualitative US-based study from 2017 on the same topic, reflecting similar clinical considerations [38]. However, US physicians frequently referenced patient satisfaction scores as a factor influencing prescribing decisions, a consideration not present in our study in the Dutch healthcare context, where such metrics are not routinely used. Furthermore, opioid prescribing in US EDs is subject to stricter regulatory frameworks, including state-level legislation that imposes limits on amounts and restricts the types of opioids that may be prescribed. In this regard, Dutch emergency physicians operate with greater clinical autonomy [1,22,38].

Previous research has shown that the risk of problematic opioid use increases with the duration of opioid prescriptions. However, even short-term use of prescription opioids carries the potential for opioid-related harms like abuse and addiction [34]. Notably, the majority of patients discharged from the ED with acute pain can be effectively managed using opioid alternatives. Specifically, appropriate dosing of acetaminophen and ibuprofen often provides sufficient analgesia for most adults [39]. So, both ED physicians and patients may benefit from education on acute pain management. It is essential to manage patient expectations by focusing on pain reduction rather than complete pain elimination [31,40]. In the US, educational programs targeting ED residents have been associated with significant reductions in opioid prescribing both in the ED and at discharge [41]. Similarly, online teaching modules have been shown to improve knowledge, confidence, and adherence to guidelines among healthcare providers regarding opioid prescriptions [42].

The majority of the participants in this study emphasize that, in addition to prescribing opioids with caution, shared decision-making is crucial. They agree that it is essential to initiate discussions about opioid use, ensuring that patients are informed about the different pain management options.

Indeed, the US Centers for Disease Control and Prevention (CDC) have issued guidelines to all healthcare providers involved in opioid management and recommend shared decision-making with patients by discussing the potential side effects, risks, and benefits of opioid use [21]. This approach has shown to not only improve patient outcomes and satisfaction but also enhances the safety and effectiveness of pain treatment by fostering a collaborative relationship between patients and healthcare providers [43]. Despite these potential benefits, participants also found that, due to time constraints in a busy ED, extensive patient education and shared decision making may not always be feasible.

## Managing problematic opioid use at the ED

EDs serve as a potential point for intervention in problematic opioid use, offering opportunities for identification, initiating treatment, and linking patients to follow-up care. Stigma surrounding substance use was recognized as a possible influential factor shaping clinicians' attitudes and prescribing decisions. Several participants noted that a history of substance use was often not explicitly explored during ED encounters, potentially reflecting discomfort around the topic. This stigma may contribute to under recognition of opioid misuse risk and missed opportunities for early intervention. Addressing this requires targeted education and communication training to normalize discussions about substance use.

Participants reported more challenges in identifying problematic opioid use, attributing these difficulties to the time constraints inherent in the ED environment and limited access to comprehensive information regarding patients' current and past medication histories.

Insight into patients' current medication use was frequently emphasized as essential for preventing "opioid shopping" and minimizing the risk of patients obtaining prescriptions from multiple sources. It was also noted that when a patient's record indicated possible opioid misuse or "shopping," clinicians sometimes preferred to consider non-opioid alternatives

first, suspecting drug-seeking behavior. While not entirely unjustified, this represents a difficult clinical and ethical balance in acute pain management.

The integration of PDMPs in the US has led to a significant decrease in opioid prescriptions with a cumulative effect of interventions like education on alternatives to opioids and modifications to electronic patient file processes [44]. In several US states, prescribers are required by law to consult the PDMP before issuing an opioid prescription. PDMPs offer a model to help healthcare providers and pharmacists identify potential drug misuse through shared data [45]. They are designed to promote safe prescribing practices and to reduce high-risk drug harms like prescription shopping, overdoses and mortality. The impact of PDMPs on reducing overdose mortality and misuse is often inconsistent and the effectiveness depends on their implementation and integration with wider public health strategies [46,47].

Adopting similar systems in the Netherlands, such as expanding access to nationwide electronic patient files, may provide a practical solution to recognize problematic prescription opioid use.

Our study participants commonly acknowledged the EDs' potential role in initiating care for patients with problematic opioid use. However, many ED physicians indicated that organizing follow-up care for patients with opioid addiction presents a significant challenge and is often perceived as impractical within the ED setting. Emergency care providers commonly reported a lack of resources and expertise to effectively manage and refer these patients. Limited collaboration with GPs and addiction specialists was identified as an issue. Additionally, as emergency care providers are accustomed to managing patients around the clock, the unavailability of GPs during after-hours was frequently highlighted as a gap in care. It is expected to worsen over time as the national shortage of GPs increases, driven by population aging, higher care demands, and limited workforce capacity. Addiction care services are generally unavailable outside regular working hours, leaving the ED as the only accessible point of contact for many patients, restricting timely access to specialized support for patients with opioid-related problems.

In the US several models have been developed to ensure continuity of care after ED discharge of patients with opioid addiction, including direct referrals to local addiction treatment services, or providing a short-term prescription for buprenorphine and referring patients to an outpatient clinic within the hospital [23,48,49]. The second option allows emergency physicians to initiate treatment and referring patients to a dedicated addiction clinic, ensuring a transition from ED care to ongoing treatment [18].

US-based studies have shown promising results from the initiation of buprenorphine and methadone in the ED for managing opioid dependency, with studies demonstrating its effectiveness in reducing overdoses and linking patients to long-term care [23,50]. Buprenorphine can be effectively used to manage opioid withdrawal and serve as a starting point for medication-assisted therapy [49]. However, participants in this study noted that this approach might be less relevant to the Netherlands due to the overall lower incidence of opioid overdoses and withdrawal cases in Dutch EDs [3].

Tailoring opioid stewardship interventions to the organization of care, such as ensuring GPs have the time, skills, and resources for patient-centered deprescribing, can help address rising trends in opioid prescriptions, hospital admissions, and mortality.

### Strengths and limitations of the study

This study offers valuable insights into the experiences and perceptions of ED physicians and PAs regarding opioid prescribing and management in the Netherlands. To our knowledge, this is the first study of its kind conducted in Europe. By including perspectives from physicians across various specialties, all working in the ED, the study captures a broad view of the challenges and opportunities in this setting. However, certain limitations must be acknowledged. Social desirability bias may have affected participants' responses, particularly on sensitive issues like opioid prescribing and addiction care. Physicians may have provided answers they perceived as more socially or professionally acceptable rather than fully disclosing their dilemmas or uncertainties. This could have led to an underestimation of the challenges they experience in practice.

 

Further, the workload of some participants impacted the interviews; busier physicians often had less time, resulting in shorter and less detailed responses compared to those who had more time to participate. This discrepancy could have influenced the findings by underrepresenting the perspectives of physicians working in high-pressure settings, where decision-making around opioid prescribing may be even more complex.

## Conclusion

This study underscores the crucial role of EDs in addressing opioid-related harms, highlighting both significant challenges and opportunities. Physicians in the ED are constantly balancing the ethical tension between relieving a patient's pain and preventing potential opioid-related harms. They often make these decisions under conditions of limited information and guidelines. As a result, everyday prescribing decisions are morally charged, requiring careful judgment about whether and how to provide opioids to patients who may be at risk for opioid use disorder. Physicians recognize the importance of early identification and intervention but emphasize the need for institutional support, clear guidelines, multidisciplinary collaboration, and effective follow-up systems to enhance their impact. Targeted interventions, including education of healthcare providers, integrated electronic patient records, and strengthened collaboration with addiction care services, are essential to bridging existing gaps. These findings contribute to ongoing efforts to refine care and policy in the face of this public health challenge.

## Supporting information

**S1 File. Checklist.**
(PDF)

**S2 File. Translate topi guide interviews.**
(PDF)

## Author contributions

**Conceptualization:** Charlotte ten Pas, Viren Bahadoer, Cornelis Kramers, Albert Dahan, Hannah Ellerbroek, Arnt Schellekens, Nicole Kraaijvanger.

**Data curation:** Charlotte ten Pas, Nicole Kraaijvanger.

**Formal analysis:** Viren Bahadoer, Hannah Ellerbroek, Arnt Schellekens, Nicole Kraaijvanger.

**Methodology:** Charlotte ten Pas, Viren Bahadoer, Hannah Ellerbroek, Nicole Kraaijvanger.

**Project administration:** Charlotte ten Pas, Joris Holkenborg, Ozcan Sir, Merel van Loon.

**Supervision:** Cornelis Kramers, Albert Dahan, Arnt Schellekens, Nicole Kraaijvanger.

**Writing – original draft:** Charlotte ten Pas.

**Writing – review & editing:** Viren Bahadoer, Cornelis Kramers, Albert Dahan, Hannah Ellerbroek, Joris Holkenborg, Ozcan Sir, Merel van Loon, Arnt Schellekens, Nicole Kraaijvanger.

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
