## [Decision Letter · Decision Letter 0]

10 Sep 2025

PLOS ONE

Dear Dr.  ten Pas,

Thank you for submitting your manuscript to PLOS ONE. After careful consideration, we feel that it has merit but does not fully meet PLOS ONE’s publication criteria as it currently stands. Therefore, we invite you to submit a revised version of the manuscript that addresses the points raised during the review process.

We look forward to receiving your revised manuscript.

Kind regards,

Isaac Amankwaa, Ph.D.

Guest Editor

PLOS ONE

Journal Requirements:

4. We note you have included a table to which you do not refer in the text of your manuscript. Please ensure that you refer to Table 2 in your text; if accepted, production will need this reference to link the reader to the Table.

Additional Editor Comments:

The most urgent revisions concern clarity and rigor in the **Methods**  (recruitment details, interviewer role, analytic procedures, and justification of data saturation) and restructuring of the **Results**  (particularly simplifying Table 2, integrating participant quotes into the text, and synthesizing themes). The **Background and Discussion**  also need strengthening: situating the Dutch findings within a broader European and U.S. context, addressing the role of guidelines, stigma, and cultural attitudes to pain, and clarifying the ethical and policy implications. In addition, consistent terminology and careful attention to phrasing are expected throughout. Please ensure your manuscript is aligned with the **COREQ checklist**  (or another relevant EQUATOR Network checklist such as SRQR).

Please revise your manuscript accordingly and submit a **point-by-point response document**  indicating how each reviewer’s comment has been addressed. Where you disagree with a suggestion, provide a clear rationale.

Reviewer's Responses to Questions

**Comments to the Author**

1. Is the manuscript technically sound, and do the data support the conclusions?

Reviewer #1: Yes

Reviewer #2: Yes

2. Has the statistical analysis been performed appropriately and rigorously?

Reviewer #1: N/A

Reviewer #2: N/A

3. Have the authors made all data underlying the findings in their manuscript fully available?

Reviewer #1: Yes

Reviewer #2: Yes

4. Is the manuscript presented in an intelligible fashion and written in standard English?

Reviewer #1: Yes

Reviewer #2: Yes

Reviewer #1: Please make the recommended grammatical and phrasing changes.

Thank you for the opportunity to review this manuscript, The role of Emergency Departments in opioid related harm: a qualitative study among emergency healthcare providers. This is the first qualitative study with ED providers in Europe and an important contribution to understanding the attitudes of providers in management opioid-related harms during an epidemic. I have made recommendations for the authors to strengthen this manuscript, including providing a more detailed Background with Netherlands-specific context that sets the stage for their recommendations, revising the format of Table 2, synthesizing their findings and shortening the presentation, and explaining how their findings are similar to/different from a comparable US-study.

Abstract

1. Line 50: I would substitute “with” for “among”

Body

1. What do you mean by, “partly due to differences in healthcare systems”? Are you saying that opioid-related harms have been constrained in the Netherlands due to differences in the HC system, compared to the US healthcare system? There seems to be an implicit comparison of these two nations. Which differences? Can you quantify them? What are examples of stricter prescribing and advertising regulations? This seems to set up your manuscript as a tension, but there is not enough information for the reader.

2. Similarly, Lines 123-124: You tell us that these are different in the Netherlands, but you haven’t told us different from whom or provided examples of these differences to illustrate.

3. Does the Netherlands have opioid prescribing guidelines for ER physicians, and what direction do they provide? Some context would be helpful for your ms.

4. Lines 87, 94, 98: You used both “opioid-related harms” and “harm”. Please use the plural form consistently in your manuscript, with “are” not “is.”

5. Line 87: I would suggest revising to: “opioid-related harms, such as overuse and dependency, addiction, and overdose (fatal and non-fatal).” I think you could also say more here about the downstream problem of diversion to street opioids after prescriptions end, and related risks of using an unsafe (adulterated) supply, as well as injection drug use risks (HIV, hep B/C, abscesses.)

6. What proportion of patients in the Netherlands who are prescribed opioids in the ED setting then develop dependency and divert to street supply? I think some quantification of “how big” this problem is, if it’s been measured, would be helpful.

7. Line 97: “Since” means “Since the event/time of.” A better choice here is “Because.”

8. Line 106: “Further, since opioid-related harm is frequently encountered in Eds.” This is confusing. What do you mean? The discussion to this point has been about post-ED prescribing opioid-related harms. What are the harms encountered in EDs? How are these harms different? Are these patients who have experienced opioid-related overdose? I see this comes a bit later (Lines 113-117), but that discussion needs to move up here, so we understand what you mean. We also need some numbers illustrating what proportion of patients are seen each year in the ED for overdoses or related treatment. In other words, what is the burden for ED physicians? Give us a sense.

9. Lines 120-121. What is problematic opioid use? This is the first time I see this term. If you are using it to refer to both existing opioid use challenges and post-ED prescribing harms, you need to state this by introducing it earlier and defining it clearly.

10. Line 122: Which guidelines? Do they exist? Do they need to be developed? Who would they be for? Same for educational initiatives. Provide context for your readers by describing the practice landscape in the Netherlands and what professional guidance and regulation currently exists.

11. Lines 147-56: We need more on recruitment. What materials or approaches did you use? Were these print? E.g. posters? Social media? Email? Listservs? How long was the recruitment period? Did participants receive any incentives? Did you have a target and did you meet it? If not, how did you decide on your sample size?

12. Lines 172-173: What difference do you think it made for a resident to conduct these interview, compared to having a non-clinician conduct them? Do you think it helped or hindered, and how? Was this a deliberate choice? Is the resident a colleague of any of these study participants? A peer, or much younger? If so, how did you handle this? I think readers might want to know how were attentive to these methodological choices, as they have power ramifications and impact the data you collect. I am not suggesting a laborious explanation here some reflection to show you are aware that these are not minor considerations.

13. Lines 178-79: insert “and” and remove the comma

14. Lines 179-190: You need more here to illustrate rigor. Did the two researchers conducting the analysis have previous experience in qualitative data analysis? Was one a senior qualitative researcher? How did they reconcile disagreements? Did they validate their codebook by test coding a sample of the interviews and comparing results? Were other team members involved in this analytical work? For example, did team members review the codes or themes?

15. Lines 200-202: Did the consent process include an explicit opt-in to quote participants?

16. Line 219: Over a quarter of your sample worked “abroad.” Can you say more? Which countries? Europe only? Or beyond?

17. Line 232, Table 2: The table is an unusual choice for presentation of findings in a qualitative study. It is unreadable. I strongly recommend you revise it. The three-level numbering is hard to understand and distracts from reading through the content. Select a model in other publications / other journals to use.

18. Line 254: “was the lack of follow-up once the patient leaves the ED.” I imagine there is a lot here that your participants discussed. Can you elaborate? This points to concerns about the structure of the health care system, possible feelings of responsibility (guilt?) of providers, and so on. I think this is rich area, if your data support exploring this in more detail.

19. Lines 257-259 and elsewhere: Quotes need to be integrated into the narrative, not placed at the bottom stand-alone. Please revise each section so that quotes are integrated into your presentation of findings, either with a colon and then indenting the quote, using “For example,…”, or sandwiching.

20. Lines 269-272: This is fascinating (and not unique to your population). Can you say more in your Discussion? What shapes differences in opinion? Training? Speciality? Age? Gender? Past experience? How do these differences become conflictual, either in patient hand offs or w/ the institution’s policies?

21. Lines 286-287: Revise so that you are not ending this sentence with a preposition. Also, why didn’t your sample always ask about a history? Is this due to lack of time, interest or curiosity, guidelines, institutional policy, training?

22. Lines 293-294: I think you need to say more about this in your Discussion. In the US, pain as the fifth vital sign, which gained acceptance through Dr. James Campbell in his 1995 presidential address to the American Pain Society, has been discredited as not evidence-based, although it is still practiced in taking a history. It is identified as a key contributor to the opioid epidemic(s) in the US, and there are excellent analyses published. It illustrates the immense influence of powerful individual physicians in shaping practice and ideology on pain management. So what is different in the Netherlands? Is this institutionalized and enshrined in practice? Has it been subject to criticism as it has in the US? Is it controversial?

23. There are too many subthemes. It leads to a cut-up presentation of findings that is overly long. Synthesize your findings thoughtfully to draw out what is relevant, and avoid presenting them in multiple choppy sections.

24. Your Discussion is quite nice and integrates the findings. Please address the questions I asked above to provide some elaboration and depth. Additionally, stigma among providers is well known and shapes prescribing attitudes. I saw it mentioned in passing once. Please elaborate on the role you think it has among your study populations and their attitudes in your Discussion.

25. While this is the first investigation of its kind in Europe, a comparable study has been published in the US, on pain management in the ED. https://pmc.ncbi.nlm.nih.gov/articles/PMC6124837/ Please incorporate this study in the Background and Discussion. How are your finding similar? What is unique to your sample and setting?

26. Lines 679-687: I think this needs to be moved to your Background, and then you can introduce the Netherlands context. Guidelines are like the ghost in the room in this manuscript: they are clearly relevant but hidden just out of sight, a kind of phantom presence. If one of your messages is that guidelines in the Netherland (and you haven’t been identified these) need to be strengthened, then be explicit about this in your Background and come back to it as a recommendation. Similarly, with PDMPs. The place to introduce these is the Background, not the Discussion. Then you can come to them in the Discussion: what do your findings suggest about these? Your narrative is confusing—I can’t tell why you’re returning to the US context when the study is in the Netherlands. You seem to suggest, rather tentatively, that the Netherlands should adopt these. Without this discussion in the Background, that (1) the Netherlands doesn’t have them, and (2) why it does not, introducing them here does not make sense. Set this up more clearly so your recommendation flows from the start. And please be confident in your recommendations.

27. Do you think that your manuscript is about ethical dilemmas in managing opioid-related harms among ED providers? The word “ethical” is not present, yet this to me is the core of what you are addressing. There is a bioethics literature that you could link to that might provide more scaffolding for your findings.

Reviewer #2: This study uses 25 semi-structured interviews with physicians and other providers in emergency departments in the Netherlands to understand the role of those departments in preventing and coping with opioid misuse, overdose, and addiction. The analysis and discussion are organized around two central themes that emerged from the analysis – one pertaining to preventing opioid-related harm, and the other pertaining to managing problematic opioid use, both within emergency departments. This is an important topic, particularly in the European context, where problematic opioid use and its harms are on the rise, and where research about how healthcare institutions can help prevent these outcomes among patients is timely. The article is well-written and the methodology is appropriate.

I am happy to recommend publication of this article, although I would recommend that the authors could make some small – but important – improvements before the manuscript is accepted. These recommendations are described in the attachment.

**Do you want your identity to be public for this peer review?** For information about this choice, including consent withdrawal, please see our Privacy Policy

Reviewer #1: **Yes: ** Janet E. Childerhose

Reviewer #2: **Yes: ** Tasleem J. Padamsee

---

## [Author Response · Author response to Decision Letter 1]

28 Oct 2025

Thank you for the thoughtful comments. We have addressed all reviewers' and editor suggestions and revised the manuscript accordingly. A detailed point-by-point response is included in the attached “Rebuttal letter” file.

---

## [Editor Report · Decision Letter 1]

23 Nov 2025

The role of Emergency Departments in opioid related harms: a qualitative study among emergency healthcare providers

PONE-D-25-24623R1

Dear Dr. Charlotte ten Pas,

We’re pleased to inform you that your manuscript has been judged scientifically suitable for publication and will be formally accepted for publication once it meets all outstanding technical requirements.

Kind regards,

Isaac Amankwaa, Ph.D.

Guest Editor

PLOS ONE
---

## [Editor Report · Acceptance letter]

PONE-D-25-24623R1

PLOS One

Dear Dr. ten Pas,

I'm pleased to inform you that your manuscript has been deemed suitable for publication in PLOS One. Congratulations! Your manuscript is now being handed over to our production team.

Kind regards,

on behalf of

Dr. Isaac Amankwaa

Guest Editor

PLOS One